# Design, Synthesis, and In Vitro/In Vivo Anti-Cancer Activities of Novel (20*S*)-10,11-Methylenedioxy-Camptothecin Heterocyclic Derivatives

**DOI:** 10.3390/ijms21228495

**Published:** 2020-11-11

**Authors:** Xiufen Dai, Guanzhao Wu, Yixuan Zhang, Xiaomin Zhang, Ruijuan Yin, Xin Qi, Jing Li, Tao Jiang

**Affiliations:** 1Key Laboratory of Marine Drugs, Ministry of Education, School of Medicine and Pharmacy, Ocean University of China, Qingdao 266003, China; fantasydxf@163.com (X.D.); wuguanzhao09@hotmail.com (G.W.); zyxlljn@163.com (Y.Z.); xiaominzhang91@163.com (X.Z.); yinruijuan@ouc.edu.cn (R.Y.); jiangtao@ouc.edu.cn (T.J.); 2Open Studio for Druggability Research of Marine Natural Products, Laboratory for Marine Drugs and Bioproducts, Qingdao National Laboratory for Marine Science and Technology, Qingdao 266237, China

**Keywords:** Camptothecin derivatives, FL118, 12e, anticancer agent, molecular design

## Abstract

A novel camptothecin analogue, (20S)-10,11-methylenedioxy-camptothecin (FL118), has been proven to show significant antitumor efficacy for a wide variety of solid tumors. However, the further development of FL118 is severely hindered due to its extremely poor water solubility and adverse side effects. Here, two series of novel 20-substituted (20S)-10,11-methylenedioxy-camptothecin coupled with 5-substituted uracils and other heterocyclic rings through glycine were synthesized. All the derivatives showed superior cytotoxic activities in vitro with IC_50_ values in the nanomolar range. Among them, 12e displayed higher cytotoxic activities in several cancer cell lines with better water solubility than FL118. Our results further showed that, like FL118, 12e inhibited cell proliferation resulting from cell cycle arrest and apoptosis by blocking the anti-apoptotic gene transcription of survivin, Mcl-1, Bcl-2, and XIAP in both A549 cells and NCI-H446 cells. Furthermore, 12e did not show any inhibitory activity on Topo I, which is involved in hematopoietic toxicity. In vivo, 12e showed similar antitumor efficacy to FL118 but lower toxicity. Our findings indicate that 12e is a promising therapeutic agent for cancer treatment, and the core structure of FL118 represents a promising platform to generate novel FL118-based antitumor drugs.

## 1. Introduction

In recent years, the incidence of malignant tumors has become a widespread public health concern. Every year, more than 10 million new cancer cases and nearly seven million deaths are reported worldwide [1]. Chemotherapy is one of the most widely used modalities for cancer treatment, with over one hundred clinically approved drugs available [2]. Among them, camptothecin (CPT) [1] (1, Figure 1), a quinoline alkaloid as an inhibitor of topoisomerase I, is a well-known drug isolated from the Chinese tree Camptotheca by Wall et al. in 1966 [3]. Over the past several decades, many CPT derivatives were synthesized and tested, and its semisynthetic analogues such as topotecan (4, Figure 1) and irinotecan (6, Figure 1) are approved by the USA Food and Drug Administration (FDA) for the treatment of various cancers, such as lung cancer [4,5,6].

In 2012, (20S)-10,11-methylenedioxy-camptothecin (FL118) (3, Figure 1), with similar structure to SN-38 (active metabolite of irinotecan) (5, Figure 1) and topotecan, was identified by using genetically modified cancer cell models via high throughput screening of chemical compound libraries [7]. Compared to other camptothecin derivatives, FL118 has been proven to possess much higher anticancer activities in many different cancer types both in vitro and in vivo [7]. Studies have demonstrated that the presence of targeted Topo I is not critical for the antitumor activity of FL118, although it has the same core structure with camptothecin [7]. In contrast, the inhibition of Topo I by FL118 may mainly be involved in the side effects of hematopoietic toxicity and possibly diarrhea as well. In fact, FL118 can function as a single anticancer drug that targets multiple different drug-resistant mechanisms. It has been reported that FL118 selectively inhibits gene promoter activities and endogenous expressions of anti-apoptotic proteins, such as survivin, XIAP, cIAP2 and Mcl-1 [8,9], contributing to cell apoptosis induction and antitumor activity [10]. Furthermore, FL118 performs those functions independent of p53 status (wild type, mutant, or null) in cells, and thus it can effectively control cancer with p53 function incapacity, in which most DNA damage drugs (if not all) show a marked lack of efficiency [7,11]. Nevertheless, further relevant research of FL118 is severely hindered due to its extremely poor water solubility and adverse side effects [10,12].

It has been revealed that anticancer activity of FL118 is highly dependent on its primary structure and steric configuration [10], suggesting that FL118 may be a promising drug platform for novel derivatives based on its core structure. In early research, it was once believed that the intact six-membered α-hydroxy lactone E-ring of CPT was indispensable for antitumor activity, because a free 20-hydroxy group favors lactone ring opening by forming intra-molecular hydrogen bonds (H-bonds), which was the main cause of failure and side effects of camptothecin [13]. However, further investigations demonstrated that modification of the free hydroxyl group at 20-position via an ester bond could be a promising way to improve antitumor efficacy in vitro and in vivo, and to reduce gastrointestinal toxicity [14,15,16,17,18,19], since esterification of the 20-hydroxy group should disfavor ring opening [20]. In this regard, esterification of 20-hydroxyl group of FL118 could eliminate the intramolecular hydrogen bonding and enhance the steric hindrance of carbonyl group of E-ring. A series of results, such as 20(S)-O-acyl esters [21], 20(S)-O-carbonate linked tripeptide conjugates [22], 20(S)-O-linked glycoconjugates [23], and 20(S)-sulfonylamidine derivatives [18,19], have proved the importance of esterified 1-derivatives as potential anticancer agents. Esterification of the 20-hydroxyl group of 1 can also function as prodrugs to release the CPTs in vivo, and enhance solubility, plasma stability, and pharmacokinetic character of CPTs [20]. Given these considerations, we designed a series of novel 5-substituted uracil and heterocycles esters of FL118 and evaluated their in vitro and in vivo antitumor activity. Results showed that these FL118 derivatives displayed impressive antitumor effects in vitro. More importantly, compound 12e showed high antitumor efficacy in both NCI-H446 and A549 xenograft mouse model without apparent toxicity, which is in line with our research expectation. Mechanically, 12e was revealed to have similar molecular mechanism to FL118 via inhibiting the transcription and expression of multiple anti-apoptotic proteins rather than Topo I activity.

## 2. Results

### 2.1. Chemistry

The chemical reactivity of 20-hydroxy group of FL118 is much lower, because this 20-hydroxy group is a tertiary alcohol with strong steric hindrance. Esterification of this hydroxy group under standard conditions is the easiest method for masking it. Therefore, we prepared prodrugs of FL118 as ester derivatives. The synthesis of FL118 was accomplished using a Friedlander condensation according to published procedures [10]. The synthetic routes to target compounds (11a–11c and 12a–12f) were outlined in Scheme 1. Briefly, the 20-hydroxy group of FL118 was converted to N-Boc protected amino esters (7) using a combination of DIC (N, N′-diisopropyl carbodiimide) and DMAP (4-dimethylaminopyridine) in a 95% yield. Subsequently, compound 7 was treated with trifluoroacetic acid in dichloromethane to afford the desired amine intermediates 8 in 2 h. Compound 9 was reacted with bromoacetic acid for 5 h to generate a carboxylic group. Condensation of the intermediates 8 with the uracil fragments by DMAP and DIC, afforded the hybrids compounds 11a–11c in 74% to 79% yields, respectively. In addition, a wide range of heterocyclic components containing pyrrole, piperidine, and substituent piperazine were all efficiently coupled with intermediates 8, which was first reacted with CDI (N, N′-Carbonyldiimidazole) as the linker, to provide compounds 12a–12f in 36% to 78% yields. All newly synthesized compounds were purified by column chromatography and their structures were confirmed by ^1^H NMR, ^13^C NMR, and HRMS.

### 2.2. 12e Inhibits the Proliferation of Various Tumor Cells

The proliferation inhibitory activities of these synthesized compounds cells were assessed by SRB assay. All the compounds displayed strong cytotoxicity on human non-small cell lung cancer cells A549 with wild type p53, the IC_50_ values were between 1.37–38.71 nM after 72 h treatment (Figure 2A). Noteworthily, 12e exhibited strongest cytotoxic activity with 10.28-fold potential than FL118. Hence, 12e was selected for further studies. Furthermore, 12e was found to have better water solubility than FL118.

The cytotoxicity of 12e on various human cancer cells is summarized in Figure 2B. It was demonstrated that 12e displayed strong inhibitory activity on these selected cancer cells with IC_50_ values in the range of 1.37–122.48 nM after 72 h of treatment. Among these cancer cells, A549 cells were most sensitive to 12e with IC_50_ value at 1.37 nM. Thus, A549 cells were used to further explore the anticancer effect and mechanism of 12e in the following research. Given that there is no new drug for small cell lung cancer nearly 30 years [24], the NCI-H446, a human small cell lung cancer line with mutant p53, was also chosen in our research. The IC_50_ value of 12e on NCI-H446 was 9.48 nM, which was very potent compared to most compounds reported.

Plate clone formation assay is the gold standard for measuring the cytotoxicity of compounds on cancer cells in vitro [25]. Accordingly, 12e showed significant inhibition on the colony-forming abilities of A549 and NCI-H446 cells in a dose-dependent manner (Figure 2C,D). Compared to the control group, the inhibition rates of 12e (0.01–1 nM) after 14 days of treatment were 22.02%, 28.81%, and 45.20% in NCI-H446 cells, 27.80%, 34.88%, and 53.41% in A549 cells, respectively. When the concentration of 12e was more than 10 nM, the colony-forming was inhibited thoroughly both in A549 and NCI-H446 cells. In order to explore whether the cell cycle arrest contributes to 12e-induced proliferation inhibition, we further analyzed the cell cycle distribution and found that 12e induced S phase arrest in A549 cells and NCI-H446 cells (Figure 2E). In the control group, the cells in S phase represented 33.86% (NCI-H446) and 39.82% (A549), and it increased to 34.95% and 47.56%.in the presence of 1 nM 12e. While in the presence of 10 and 100 nM 12e, the cells in S phase rose to 100% (Figure 2F). These results indicate that 12e suppresses the proliferation of lung cancer cells by arresting the cell cycle in the S phase, which is similar to effect of FL118 on the cell cycle [26,27].

### 2.3. 12e Induces Cell Apoptosis of A549 Cells and NCI-H446 Cells

Next, we determine whether S phase arrest induced by 12e resulted in cell apoptosis. In apoptosis, the nucleus undergoes a series of changes including segregation of nucleoli and condensation of chromatin [28]. The nuclear morphology was evaluated after Hoechst 33342 staining. As shown in Figure 3A, the number of cell nuclei that exhibited brighter blue fluorescence, shrinkage, or DNA fragmentation (shown by arrows) was increased significantly after exposure to 12e in A549 cells and NCI-H446 cells dose-dependently. Furthermore, the levels of γ-H2AX, an essential marker of DNA double-strand breaks, was increased significantly in a dose-dependent manner (Figure 3B), suggesting that 12e induced DNA double-strand breaks, which was consistent with the influence of FL118 on DNA strand [29].

Cysteinyl aspartate specific proteinase-9 (caspase-9) is the apical caspases in the intrinsic apoptosis pathways, and cysteinyl aspartate specific proteinase-3 (caspase-3) is considered to be the most important component to the effectors caspases. The cleaving and activation of the caspase-9/caspase-3 is a hallmark of the intrinsic apoptosis [30]. In addition, the poly-adenosine diphosphate-ribose polymerase (PARP) protein is a substrate of caspase-3. Cleaved PARP seems to be an early marker of apoptosis in cells. Bax, a member of the Bcl-2 family, can promote apoptosis mediated by mitochondria. As shown in Figure 3B, we observed obvious increases of cleaved caspase-9, cleaved caspase-3, cleaved PARP, and pro-apoptotic protein bax in the presence of 12e. Meanwhile, the levels of caspase-8, an important component of the death signal receptor pathway of apoptosis [31], had no change after exposure to 12e treatment. Taken together, 12e induces A549 and NCI-H446 cell apoptosis involved in mitochondrial pathway rather than death receptor pathway.

In order to quantify the extent of apoptosis, flow cytometry analysis was performed to detect 12e-induced apoptosis rates by Annexin V-FITC and PI staining. Annexin V-FITC and PI double staining can distinguish apoptotic cells, necrotic cells as well as normal cells. The specific binding of phosphatidylserine exposed on the cell surfaces undergoing apoptosis with Annexin V can be used to detect apoptotic cells. The late apoptotic cells and necrotic cells were assessed by propidium iodide (PI) staining [32]. As shown in Figure 3C, after treatment by 12e (0, 1, 10, and 100 nM) for 24 h, the rates of early and late apoptosis (AV+/PI-) were 6.76%, 12.57%, 18.52%, and 20.32% in NCI-H446 cells, and 6.84%, 12.55%, 15.61%, and 24.17% in A549 cells, respectively. These results support that 12e induces cell apoptosis.

### 2.4. 12e does not Impair Cell Proliferation Signaling in A549 Cells and NCI-H446 Cells

Cell proliferations, apoptosis and differentiations are regulated by cell signal transduction pathway. In various cancer cells, stat3, AKT and erk1/2 were abnormally activated, which are molecules in three main signaling pathway, relating to the cell proliferation and survive [33,34,35]. To investigate which cell signaling transduction pathway was involved in 12e-induced cell apoptosis, we investigated the expressions and activations of stat3, AKT and erk1/2 by western blot assay. As shown in Figure 4, after treatment with 1, 10, and 100 nM 12e for 24 h, the levels of stat3, AKT, and erk1/2 had no significant change, and the phosphorylation levels of stat3, AKT, and erk1/2 were not appreciably decreased on NCI-H446 and A549 cells. These results indicate that 12e-induced apoptosis is probably not associated with the inactivation of cell proliferation upstream signaling.

### 2.5. 12e Reduces the Transcription and Expression of Multiple Anti-Apoptotic Proteins

The IAP family and Bcl-2 family proteins play important roles in suppressing apoptosis of cells [36]. It was reported that FL118 inhibited the transcriptions and expressions of survivin, XIAP and Mcl-1 [8,9]. We then tested whether 12e affected the expression levels of these anti-apoptotic proteins. After treatment with 1 to 100 nM of 12e for 24 h, the protein levels of survivin, Bcl-2, Mcl-1, and XIAP were down-regulated both in A549 and NCI-H446 cells in a dose-dependent manner (Figure 5A). Accordingly, the RT-PCR assay showed that the mRNA levels of survivin, Bcl-2, Mcl-1 and XIAP were decreased significantly (Figure 5B). All these data suggest that 12e inhibits the genetic transcription of multiple anti-apoptotic proteins and their subsequent expression. Early studies have demonstrated that camptothecin (CPT) inhibits the synthesis of DNA and protein, due to inhibition of DNA topoisomerase I (Topo I) activity [37]. Previous studies showed that the inhibition of Topo I activity by FL118 did not play a major role in its antitumor activity [7,10]. To determine whether 12e inhibited DNA Topo I activity, a Topo I -Mediated Relaxation of pBR322 DNA Assay was performed. The pBR322 DNA was presented as supercoiled state (SC) without treatment of Topo I, and relaxed state (RLX) with the treatment of Topo I. After co-treated with CPT, the relaxation activity of Topo I was inhibited and SC DNA was reversely increased distinctly, while the pBR322 DNA was still presented as RLX after co-treated with FL118, or 12e (Figure 5C). Intriguingly, FL118 exhibited a bit weak inhibitory activity on Topo I than 12e, as 12e did not show any inhibitory activity on it. Meanwhile, we observed that compounds 12a and 12c showed no inhibitory activity on Topo I. These results suggest that 12e inhibits transcriptions and expressions of multiple anti-apoptotic proteins, thereby inducing cell apoptosis of A549 and NCI-H446 cells independent of Topo I activity.

### 2.6. 12e Suppresses the Growth of Lung Cancer Cells Xenograft In Vivo

Next, we studied the in vivo antitumor activity of 12e using animal xenograft models of human A549 and NCI-H446 cancer cells. We compared the antitumor activities of 12e and FL118 at a dose of 10 mg/kg by intragastric administration once per week. As shown in Figure 6, 12e exhibited a distinct reduction in tumor volume compared to vehicle in A549 and NCI-H446 xenografts with similar antitumor activity to FL118 (Figure 6A,B). The inhibitory rates of 12e and FL118 were 48.8% and 59.8% (A549 xenografts), and 70.3% and 75.4% (NCI-H446 xenografts) respectively. During the period of drug administration, the mice body weights of FL118 group lost obviously both in A549 and NCI-H446 xenograft models compared with control group, while 12e did not induce an obvious loss of mice body weight (Figure 6C,D). The results indicated that 12e has weaker toxicity than FL118. Furthermore, just like FL118 12e down-regulated the expression of survivin, Mcl-1, Bcl-2 and XIAP in A549 and NCI-H446 xenograft tumors (Figure 6E). These findings indicate that 12e suppresses lung cancer growth in vivo through same mechanism as FL118, but with weak toxicity.

## 3. Discussion

In this study, novel 20-substituted FL118 derivatives coupled with 5-substituted uracils and other heterocyclic rings through glycine were synthesized. All the derivatives showed superior cytotoxic activities in vitro, with IC_50_ values between 1.37–38.71 nM after 72 h treatment in A549 cell lines. Among them, 12e displayed higher cytotoxic activities on various human cancer cells including A549 and NCI-H446 cells. In addition, 12e was found to have better water solubility and more potential cytotoxic effect than FL118. Cell cycle arrest and apoptosis were the two main causes of growth inhibition [38]. Moreover, 12e suppressed the proliferation of NCI-H446 and A549 cells by arresting the cell cycle at the S phase, which resulted in cell apoptosis. This result is consistent with previous studies that FL118 induced S phase arrest in several types of human cancer cell line [19,20]. With Hoechst 33342 staining, western blot and flow cytometry analysis, our data showed that 12e induced apoptosis in NCI-H446 and A549 lung cells via up-regulation of γH2AX and bax, down-regulation of survivin, Bcl-2, Mcl-1 and XIAP with cleavage and activation of caspase-3, caspase-9, and PARP. PI3K/AKT, JAK/STAT3, and RAS/ERK1/2 signaling pathways contribute to a variety of processes that are critical in mediating many aspects of cellular function including cell growth and survival, abnormalities of these pathways have been reported in many human tumors, and inhibitors against these pathways promote cell apoptosis [39]. However, there is no change in the activations and expressions of AKT, stat3 and erk1/2 after 12e treatment, indicating that 12e plays a role in the downstream of these signaling pathway related to cell growth and survival. As expected, 12e was verified to inhibit the genetic transcription of multiple anti-apoptotic proteins, which is in line with the report that FL118 binds to similar promoter region of the survivin, Mcl-1, Bcl-2 and XIAP genes, and thereby inhibiting these genes transcription. Moreover, we confirmed that FL118 exhibited weaker inhibitory activity on Topo I than 12e, as 12e did not show any inhibitory activity on Topo I. Inhibition of Topo I by FL118 is predominantly involved in hematopoietic toxicity [25], but not in FL118 antitumor activity, which may be an explanation for the lower toxicity of 12e. Previous studies indicated that FL118 treatment in its oral formulation has a maximum tolerance dose of 10 mg/kg in human tumor-bearing SCID with the weekly × 4 schedule [19]. Similarly, we found that 12e showed high antitumor efficacy in both NCI-H446 and A549 mouse models at a dose of 10 mg/kg with the weekly ×4 schedule. More importantly, 12e did not cause apparent toxicity in xenograft mice compared to FL118, which is in line with our original intention of molecular design.

In summary, through esterification of the 20-hydroxy group of FL118 with 5-substituted uracils and N-containing heterocycles via a glycine linker, we successfully obtained a new compound 12e with similar antitumor activity to FL118 while lower toxicity. In addition, mechanism research reveals that 12e inhibits transcriptions and expressions of anti-apoptotic protein survivin, Bcl-2, Mcl-1 and XIAP in line with molecular mechanism of FL118. Our findings indicate that the core structure of FL118 represents a promising platform for the generation of novel FL118 derivatives, which lays the foundation for further antitumor drug development in the future.

## 4. Materials and Methods

### 4.1. Chemistry

#### 4.1.1. Materials and Methods

All reagents and solvents were reagent grade or were purified by standard methods before use. Melting points were determined in open capillaries and are uncorrected. ^1^H and ^13^C NMR spectra were recorded at 400 MHz and 100 MHz on a Bruker AM-400 spectrometer using TMS as reference ( Bruker Daltonics Inc., Billerica, MA, USA). Chemical shifts (δ values) and coupling constants (*J* values) are given in ppm and Hz, respectively. Anhydrous tetrahydrofuran (THF) was obtained by distillation from sodium-benzophenone ketyl; dry methylene chloride was obtained by distillation from phosphorus pentoxide. All reactions requiring anhydrous conditions were performed under a positive nitrogen flow, and all glassware were oven dried and/or flame dried. Isolation and purification of the compounds were performed by flash column chromatography on silica gel (200–300 mesh) produced by Qingdao Marine Chemical Factory, Qingdao (China). Analytical thin-layer chromatography (TLC) was conducted on Fluka TLC plates (silica gel 60 F254, aluminum foil).

#### 4.1.2. Synthesis of FL118

FL118 was synthesized according to our previous procedures [10], namely in three steps from the commercially available piperonal.

#### 4.1.3. Synthesis of (S)-7-ethyl-8,11-dioxo-7,8,11,13-tetrahydro-10H-[1,3]dioxolo[4,5-g]pyrano[3′,4′:6,7] indolizino[1,2-b]quinolin-7-yl (tert-butoxycarbonyl)glycinate (7)

FL118 (0.50 mmol), Boc-Gly-OH (0.60 mmol), and DMAP (1.02 mmol) were dissolved in anhydrous DCM (50 mL) under N_2_ at 0 °C. DIC (1.20 mmol) was added dropwise into the reaction mixture. The mixture was stirred for 1 h at 0 °C and then stirred overnight at room temperature. The mixture was washed with 100 mM HCl aqueous solution (3 × 100 mL), water (3 × 100 mL), and saturated brine (3 × 100 mL). The organic layer was dried over anhydrous Na_2_SO_4_ and purified by silica gel column chromatography (Acetone/DCM = 1:20 to 10:1, *v*/*v*) (95% yield). mp 241–242 °C; HRMS (ESI): calcd for C_28_H_27_N_3_O_9_ 550.5360, found 550.1806. ^1^H NMR (500 MHz, DMSO) δ 8.34 (s, 1H), 7.36 (s, 2H), 7.02 (d, *J* = 70.1 Hz, 1H), 6.24 (s, 2H), 5.45 (s, 2H), 5.08 (d, *J* = 25.9 Hz, 2H), 3.87 (dd, *J* = 54.0, 17.3 Hz, 2H), 2.10 (s, 2H), 1.32 (d, 6H), 0.94 (d, 6H). ^13^C NMR (125 MHz, DMSO) δ 169.98, 167.64, 157.03, 156.29, 151.82, 149.97, 149.09, 146.81, 146.63, 145.88, 130.60, 128.58, 125.99, 118.23, 104.79, 103.46, 103.07, 95.30, 79.08, 76.70, 66.61, 50.46, 42.34, 41.06, 30.69, 28.20, 23.63, 7.93.

#### 4.1.4. Synthesis of (S)-7-ethyl-8,11-dioxo-7,8,11,13-tetrahydro-10H-[1,3]dioxolo[4,5-g]pyrano[3′,4′:6,7] indolizino[1,2-b]quinolin-7-yl glycinate (8)

Compound 7 (0.36 mmol) was dissolved in TFA (20 mL), and the solution was stirred for 1 h at room temperature. The solution was evaporated and purified by silica gel column chromatography (Acetone/DCM = 1:10 to 1:1, *v*/*v*) (95% yield). mp > 250 °C; HRMS (ESI): calcd for C_23_H_19_N_3_O_7_ 450.4190, found 450.1288. ^1^H NMR (500 MHz, DMSO) δ 8.44 (s, 1H), 7.45 (s, 1H), 7.40 (s, 1H), 7.16 (s, 1H), 6.25 (s, 2H), 5.48 (d, *J* = 16.5 Hz, 2H), 5.16 (d, *J* = 20.9 Hz, 2H), 4.26 (d, *J* = 17.9 Hz, 1H), 4.06 (d, *J* = 17.9 Hz, 1H), 2.21 – 2.09 (m, 2H), 0.98 − 0.89 (m, 3H). ^13^C NMR (125 MHz, DMSO) δ 167.34, 167.15, 156.99, 151.97, 149.97, 149.22, 146.89, 146.86, 145.20, 130.85, 128.80, 126.18, 118.33, 104.77, 103.62, 103.12, 95.17, 78.01, 66.75, 50.61, 30.57, 23.63, 7.93.

#### 4.1.5. General Procedure for the Synthesis of Compounds 11a–11c

Compound **8** (0.13 mmol), 5-substituted uracil acid 10a–10c (1.1 mmol), and DMAP (0.20 mmol) were dissolved in anhydrous DCM (20 mL) at 0 °C. DIC (0.20 mmol) was added dropwise into the reaction mixture. The mixture was stirred for 1 h at 0 °C and then stirred for overnight at room temperature. The mixture was washed with 100 mM HCl aqueous solution (3 × 100 mL), water (3 × 100 mL), and saturated brine (3 × 100 mL). The organic layer was dried over anhydrous Na_2_SO**_4_** and purified by silica gel column chromatography (Acetone/DCM = 1:20 to 10:1, *v*/*v*) to yield the desired compound, 11a–11c, respectively.

##### (S)-7-ethyl-8,11-dioxo-7,8,11,13-tetrahydro-10H-[1,3]dioxolo[4,5-g]pyrano[3′,4′:6,7] indolizino[1,2-b]quinolin-7-yl (2-(2,4-dioxo-3,4-dihydropyrimidin-1(2H)-yl)acetyl)glycinate (11a)

The general synthetic method described above afforded compound 11a as a yellow solid (77%). mp > 250 °C; HRMS (ESI): calcd for C_29_H_23_N_5_O_10_ 602.5280, found 602.1506. ^1^H NMR (500 MHz, DMSO) δ 11.21 (s, 1H), 8.71 (s, 1H), 8.38 (s, 1H), 7.46 (d, *J* = 31.5 Hz, 3H), 7.01 (s, 1H), 6.26 (s, 2H), 5.46 (s, 3H), 5.11 (s, 2H), 4.35 (s, 2H), 4.22 (s, 1H), 4.09 (s, 1H), 2.15 (s, 2H), 0.89 (s, 3H). ^13^C NMR (125 MHz, DMSO) δ 169.16, 167.91, 167.52, 164.22, 156.91, 151.84, 151.37, 150.10, 149.12, 146.86, 145.38, 130.58, 128.75, 127.84, 126.08, 118.48, 114.29, 105.07, 103.50, 103.05, 101.07, 94.70, 76.84, 66.79, 50.51, 49.67, 30.86, 7.97.

##### (S)-7-ethyl-8,11-dioxo-7,8,11,13-tetrahydro-10H-[1,3]dioxolo[4,5-g]pyrano[3′,4′:6,7]indolizino[1,2-b]quinolin-7-yl (2-(5-chloro-2,4-dioxo-3,4-dihydropyrimidin-1(2H)-yl)acetyl)glycinate (11b)

The general synthetic method described above afforded compound **11b** as a yellow solid (79%), mp 242-243 °C; HRMS (ESI): calcd for C_29_H_22_ClN_5_O_10_ 636.9700, found 636.1151. ^1^H NMR (500 MHz, DMSO) δ 11.79 (s, 1H), 8.74 (s, 1H), 8.38 (s, 1H), 8.03 (s, 1H), 7.42 (s, 2H), 7.01 (s, 1H), 6.26 (s, 2H), 5.46 (s, 2H), 5.11 (s, 2H), 4.36 (s, 2H), 4.24 (dd, *J* = 18.3, 5.5 Hz, 1H), 4.09 (dd, *J* = 17.9, 5.3 Hz, 1H), 2.19 − 2.09 (m, 2H), 0.90 (*t*, *J* = 7.1 Hz, 3H). ^13^C NMR (125 MHz, DMSO) δ 169.11, 167.61, 167.51, 159.95, 156.91, 151.84, 150.51, 150.09, 149.12, 146.90, 146.86, 145.39, 144.01, 130.55, 128.74, 126.07, 118.45, 106.38, 105.08, 103.48, 103.04, 94.70, 76.86, 66.79, 50.50, 49.99, 40.72, 30.85, 7.97.

##### (S)-7-ethyl-8,11-dioxo-7,8,11,13-tetrahydro-10H-[1,3]dioxolo[4,5-g]pyrano[3′,4′:6,7]indolizino[1,2-b]quinolin-7-yl (2-(5-fluoro-2,4-dioxo-3,4-dihydropyrimidin-1(2H)-yl)acetyl)glycinate (11c)

The general synthetic method described above afforded compound 11c as a yellow solid (74%), mp 242-243 °C; HRMS (ESI): calcd for C_29_H_22_FN_5_O_10_ 620.5184, found 620.1409. ^1^H NMR (500 MHz, DMSO) δ 8.73 (s, 1H), 8.37 (s, 1H), 7.97 (d, *J* = 6.4 Hz, 1H), 7.41 (s, 2H), 7.01 (s, 1H), 6.26 (s, 2H), 5.46 (s, 2H), 5.09 (s, 2H), 4.29 (d, *J* = 18.1 Hz, 2H), 4.26 − 4.17 (m, 1H), 4.11 − 4.02 (m, 1H), 2.19 − 2.07 (m, 2H), 0.90 (*t*, *J* = 6.8 Hz, 3H). ^13^C NMR (125 MHz, DMSO) δ 169.12, 167.68, 167.51, 156.91, 151.84, 150.07, 150.04, 149.11, 146.90, 146.84, 145.40, 140.65, 131.35, 131.07, 130.54, 128.70, 126.05, 118.46, 105.04, 103.47, 103.05, 94.71, 76.85, 66.79, 50.48, 49.86, 40.72, 30.86, 7.97.

#### 4.1.6. General Procedure for the Synthesis of Compounds 12a–12f

Compound **8** (0.11 mmol) and Triethylamine (0.02 mmol) was dissolved in anhydrous DCM (50 mL) under N_2_ at 0 °C. Then CDI (0.14 mmol) in anhydrous DCM (10 mL) was then added. After stirring at room temperature for 1 h, various heterocycles (0.154 mmol) were added into the solution, and the reaction mixture was stirred overnight at room temperature. The mixture was washed with water (3 × 100 mL), and saturated brine (3 × 100 mL). The organic layer was dried over anhydrous Na_2_SO_4_ and purified by silica gel column chromatography (Acetone/DCM = 1:8 to 5:1, *v*/*v*) to yield the desired compound, 12a–12f, respectively.

##### (S)-7-ethyl-8,11-dioxo-7,8,11,13-tetrahydro-10H-[1,3]dioxolo[4,5-g]pyrano[3′,4′:6,7]indolizino[1,2-b]quinolin-7-yl (morpholine-4-carbonyl)glycinate (12a)

The general synthetic method described above afforded compound 12a as a yellow solid (78%). mp > 250 °C; HRMS (ESI): calcd for C_28_H_26_N_4_O_9_ 563.5350, found 563.1758. ^1^H NMR (500 MHz, DMSO) δ 8.42 (s, 1H), 7.46 (s, 1H), 7.42 (s, 1H), 7.14 (*t*, *J* = 5.7 Hz, 1H), 7.05 (s, 1H), 6.27 (d, *J* = 4.8 Hz, 2H), 5.45 (s, 2H), 5.15 (s, 2H), 4.02 (dd, *J* = 17.7, 5.7 Hz, 1H), 3.90 (dd, *J* = 17.8, 5.8 Hz, 1H), 3.48 (t, *J* = 4.7 Hz, 4H), 3.25 (dt, *J* = 13.5, 6.6 Hz, 4H), 2.12 (td, *J* = 13.8, 7.0 Hz, 2H), 0.89 (*t*, *J* = 7.3 Hz, 3H). ^13^C NMR (125 MHz, DMSO) δ 170.50, 167.64, 157.76, 156.94, 151.85, 150.20, 149.12, 146.92, 146.68, 145.54, 130.58, 128.76, 126.05, 118.48, 105.01, 103.53, 103.06, 95.08, 76.45, 66.76, 66.35, 50.49, 44.21, 42.44, 30.93, 7.93.

##### (S)-7-ethyl-8,11-dioxo-7,8,11,13-tetrahydro-10H-[1,3]dioxolo[4,5-g]pyrano[3′,4′:6,7]indolizino[1,2-b]quinolin-7-yl (piperidine-1-carbonyl)glycinate (12b)

The general synthetic method described above afforded compound 12b as a yellow solid (76%), mp >250 °C; HRMS (ESI): calcd for C_29_H_28_N_4_O_8_ 560.5634, found 560.1875. ^1^H NMR (500 MHz, DMSO) δ 8.42 (s, 1H), 7.46 (s, 1H), 7.42 (s, 1H), 7.14 (*t*, *J* = 5.7 Hz, 1H), 7.05 (s, 1H), 6.27 (d, *J* = 4.8 Hz, 2H), 5.45 (s, 2H), 5.15 (s, 2H), 4.02 (dd, *J* = 17.7, 5.7 Hz, 1H), 3.90 (dd, *J* = 17.8, 5.8 Hz, 1H), 3.48 (*t*, *J* = 4.7 Hz, 4H), 3.25 (dt, *J* = 13.5, 6.6 Hz, 4H), 2.12 (td, *J* = 13.8, 7.0 Hz, 2H), 0.89 (*t*, *J* = 7.3 Hz, 3H). ^13^C NMR (125 MHz, DMSO) δ 172.96, 170.72, 167.68, 157.47, 156.95, 151.81, 150.21, 149.06, 146.63, 145.65, 130.58, 128.71, 126.01, 118.43, 105.23, 103.52, 96.32, 95.26, 76.38, 72.85, 66.73, 65.67, 50.58, 44.69, 42.68, 30.66, 25.80, 24.51, 8.24.

##### (S)-7-ethyl-8,11-dioxo-7,8,11,13-tetrahydro-10H-[1,3]dioxolo[4,5-g]pyrano[3′,4′:6,7]indolizino[1,2-b]quinolin-7-yl (pyrrolidine-1-carbonyl)glycinate (12c)

The general synthetic method described above afforded compound 12c as a yellow solid (76%), mp > 250 °C; HRMS (ESI): calcd for C_28_H_26_N_4_O_8_ 547.5360, found 547.1809. ^1^H NMR (500 MHz, DMSO) δ 8.34 (d, *J* = 3.2 Hz, 1H), 7.36 (s, 1H), 7.17 (*t*, *J* = 5.8 Hz, 1H), 7.13 (d, *J* = 8.0 Hz, 1H), 6.21 (d, *J* = 2.3 Hz, 2H), 5.41 (dd, *J* = 30.3, 16.6 Hz, 2H), 4.94 (dd, *J* = 22.9, 12.8 Hz, 2H), 4.02 (dd, *J* = 17.7, 5.7 Hz, 1H), 3.96 − 3.83 (m, 1H), 3.71 (d, *J* = 14.6 Hz, 1H), 3.25 (dt, *J* = 13.5, 6.6 Hz, 4H), 2.12 (td, *J* = 13.8, 7.0 Hz, 2H), 1.87 − 1.79 (m, 1H), 1.53 − 1.30 (m, 5H), 0.98 (dd, *J* = 21.3, 6.8 Hz, 3H). ^13^C NMR (125 MHz, DMSO) δ 170.76, 167.64, 156.97, 156.57, 151.86, 150.24, 149.16, 146.97, 146.75, 145.45, 135.04, 130.65, 128.86, 126.13, 120.43, 118.63, 105.09, 103.61, 103.06, 95.00, 76.34, 66.82, 50.56, 45.72, 42.16, 31.04, 25.45, 7.89.

##### (S)-7-ethyl-8,11-dioxo-7,8,11,13-tetrahydro-10H-[1,3]dioxolo[4,5-g]pyrano[3′,4′:6,7]indolizino[1,2-b]quinolin-7-yl (4-methylpiperazine-1-carbonyl)glycinate (12d)

The general synthetic method described above afforded compound 12d as a yellow solid (36%), mp 243-244 °C; HRMS (ESI): calcd for C_29_H_29_N_5_O_8_ 576.5780, found 576.2073. ^1^H NMR (500 MHz, DMSO) δ 8.36 (s, 1H), 7.37 (s, 1H), 7.34 (s, 1H), 7.10 (s, 1H), 6.21 (d, *J* = 2.3 Hz, 2H), 5.42 (q, *J* = 16.8 Hz, 2H), 5.05 (s, 2H), 4.03 (d, *J* = 17.9 Hz, 1H), 3.92 (d, *J* = 17.9 Hz, 1H), 3.36 (s, 4H), 2.59 (s, 4H), 2.36 (s, 3H), 2.08 (dt, *J* = 24.0, 7.0 Hz, 2H), 0.89 (*t*, *J* = 7.3 Hz, 3H). ^13^C NMR (125 MHz, DMSO) δ 170.45, 167.84, 157.51, 157.15, 151.99, 149.78, 149.17, 146.67, 146.48, 146.10, 130.94, 128.51, 126.09, 118.22, 104.60, 103.48, 103.11, 66.58, 53.86, 53.63, 45.17, 44.38, 42.33, 42.24, 30.70, 7.86.

##### (S)-7-ethyl-8,11-dioxo-7,8,11,13-tetrahydro-10H-[1,3]dioxolo[4,5-g]pyrano[3′,4′:6,7]indolizino[1,2-b]quinolin-7-yl (2-methylpiperidine-1-carbonyl)glycinate (12e)

The general synthetic method described above afforded compound 12e as a yellow solid (78%), mp 240–242 °C; HRMS (ESI): calcd for C_30_H_30_N_4_O_8_ 575.5900, found 575.2121. ^1^H NMR (500 MHz, DMSO) δ 8.28 (d, *J* = 3.2 Hz, 1H), 7.32 – 7.28 (m, 1H), 7.25 (d, *J* = 10.6 Hz, 1H), 7.13 (d, *J* = 8.0 Hz, 1H), 6.20 (d, *J* = 5.0 Hz, 2H), 5.41 (dd, *J* = 30.3, 16.6 Hz, 2H), 4.94 (dd, *J* = 22.9, 12.8 Hz, 2H), 4.07 − 3.97 (m, 1H), 3.97 − 3.85 (m, 1H), 3.71 (d, *J* = 14.6 Hz, 1H), 2.69 (dt, *J* = 26.3, 12.7 Hz, 1H), 2.14 − 2.01 (m, 2H), 1.87 − 1.79 (m, 1H), 1.53 − 1.30 (m, 5H), 0.98 (dd, *J* = 21.3, 6.8 Hz, 3H), 0.93 − 0.83 (m, 3H). ^13^C NMR (125 MHz, DMSO) δ 172.97, 170.74, 157.28, 156.98, 151.79, 150.51, 150.40, 149.09, 147.02, 146.36, 130.61, 128.86, 126.05, 118.53, 105.27, 103.56, 103.03, 96.34, 76.40, 72.86, 65.69, 55.37, 50.62, 45.60, 38.35, 30.68, 18.75, 15.74, 8.24.

##### (S)-7-ethyl-8,11-dioxo-7,8,11,13-tetrahydro-10H-[1,3]dioxolo[4,5-g]pyrano[3′,4′:6,7]indolizino[1,2-b]quinolin-7-yl (thiomorpholine-4-carbonyl)glycinate (12f)

The general synthetic method described above afforded compound 12f as a yellow solid (70%), mp > 250 °C; HRMS (ESI): calcd for C_28_H_26_N_4_O_8_S 579.5860, found 579.1544. ^1^H NMR (500 MHz, DMSO) δ 8.43 (s, 1H), 7.47 (s, 1H), 7.42 (s, 1H), 7.17 (*t*, *J* = 5.8 Hz, 1H), 7.09 (d, *J* = 5.9 Hz, 1H), 6.27 (d, *J* = 4.9 Hz, 2H), 5.46 (s, 2H), 5.17 (d, *J* = 9.6 Hz, 2H), 4.01 (dd, *J* = 17.7, 5.7 Hz, 1H), 3.89 (dd, *J* = 17.7, 5.7 Hz, 1H), 3.59 (ddd, *J* = 12.9, 9.1, 4.3 Hz, 3H), 3.29 − 3.25 (m, 1H), 2.84 − 2.80 (m, 1H), 2.46 (t, *J* = 4.9 Hz, 3H), 2.13 (tt, *J* = 14.2, 7.2 Hz, 2H), 0.91 (*t*, *J* = 7.4 Hz, 3H). ^13^C NMR (125 MHz, DMSO) δ 170.56, 167.66, 157.18, 156.97, 151.85, 150.23, 149.13, 146.94, 146.69, 145.65, 130.58, 128.78, 126.05, 118.42, 105.03, 103.55, 103.06, 95.20, 76.49, 66.74, 50.52, 46.54, 45.21, 42.70, 30.91, 26.55, 24.08, 7.97.

### 4.2. Biological Evalutation

#### 4.2.1. Cell Culture

Human colorectal cancer cell lines HCT-8, gastric cancer cell line MGC80-3, leukemia cell lines HL-60 and K562, pancreatic cancer cell line ASPC-1 and PNAC-1, hepatoma cell line HepG2, neuroglioma cell line U87, neuroblastoma cell line SH-SY5Y, NSCLC cell line A549, and SCLC cell line NCI-H446 were provided by the Institute of Biochemistry and Cell Biology, Chinese Academy of Sciences (Shanghai, China). NCI-H446, HCT-8, MGC80-3, and ASPC-1 were cultured in 1640 medium (Gibco, Grand Island, NY, USA), which were supplemented with 10% FBS (Gibco, Grand Island, NY, USA), additional 25 mg/L glucose and 1 mM sodium pyruvate (Solarbio, Beijing, China). HL-60 and HepG2 were maintained in 1640 supplemented with 10% FBS. A549 cells were maintained in F12K medium (Gino, Hangzhou, China), which were supplemented with 10% FBS. K562 cells were maintained in IMDM medium (Gibco, Grand Island, NY, USA) supplemented with 10% FBS. PNAC-1 cells were maintained in DMEM medium (Gibco, Grand Island, NY, USA) supplemented with 10% FBS. All cells were maintained in 5% CO_2_ at 37 °C and passaged every 2–4 days.

#### 4.2.2. Cell Proliferation Assay

Cell proliferation was measured by the SRB (Sigma, St. Louis, MO, USA) or 3-(4,5-Dimethylthiazol-2-yl)-2,5-Diphenyltetrazolium Bromide reagent (MTT, Solarbio, Beijing, China) assay. Cells were seeded into 96-well plates, and treated with various concentrations of the indicated samples for 72 h. MTT (10 μL, 5 mg/mL) was added and incubated at 37 °C for another 4 h. The formazan product was dissolved and quantitated by spectrophotometry at a wavelength of 570 nm. Inhibition rate of each sample was calculated from the A570 nm values as follows: Inhibition rate (%) = (A570 nm control − A570 nm sample)/A570 nm control ×100%. For SRB assay, the cells are immobilized with TCA and tinted with SRB stain. The SRB stain that bound to cells was dissolved and quantitatively analyzed at a wavelength of 515 nm. The cytotoxicity of compounds was expressed as IC_50_. All experiments were repeated at least three times.

#### 4.2.3. Hoechst 33342 Staining

NCI-H446 cells and A549 cells were treated with 12e (0–100 Nm) for 24 h, then were stained with Hoechst 33342 (5 μg/mL, Sigma, St. Louis, MO, USA). The nuclear morphology was taken photos under fluorescence microscope.

#### 4.2.4. Clonogenic Assay

NCI-H446 or A549 cells were seeded at 500 cells/well in 6-well plates, treated with various concentrations of 12e for 14 days. Fixed by methanol at room temperature and stained with Giemsa (Sigma, St. Louis, MO, USA) solution, colonies were defined as ≥50 cells/colony. Finally, colonies were counted and photographed.

#### 4.2.5. Annexin V-FITC/PI Double-Staining Assay

The apoptosis was performed using the Annexin V-FITC/PI apoptosis detection kit (Absin Bioscience Inc., Shanghai, China). Briefly, cells (3 × 10^5^) were incubated with 12e (0–100 nM) for 24 h, and then harvested by centrifugation, washed with ice-cold PBS twice, and resuspended in binding buffer. Staining was started by adding Annexin V-FITC (5 μL) and PI (5 μL) followed by incubation for 10 min at room temperature in the dark. Then, samples were immediately analyzed by NovoCyte flow cytometer (ACEA Biosciences, San Diego, CA, USA).

#### 4.2.6. Cell Cycle Analysis

NCI-H446 and A549 cells were incubated with 12e (0–100 nM) for 24h. After that, the cells were collected and washed in PBS and fixed in ice-cold 70% (*v*/*v*) ethanol overnight at −20 °C. The cell pellet was resuspended in PBS and stained with a mixture of RNase (10 μg/mL, Solarbio, Beijing, China) and PI (50 μg/mL, Sigma, St. Louis, MO, USA) for 20 min in the dark. Cell cycle distribution analysis was performed using a MoFlo XDP flow cytometry system (Beckman Coulter, Boulevard Brea, CA, USA).

#### 4.2.7. Western Blotting

Cells were incubated with various concentrations of 12e for 24 h and then washed twice in cold phosphate buffered saline (PBS). Cells were lysed with lysis buffer (10 mM Tris, pH 7.4, 150 mM NaCl, 1 mM ethylenediaminetetraacetic acid (EDTA), 1% Triton X-100, 0.5% NP-40, 1 mM propidium iodide (PI), 1mM dithiothreitol (DTT), 1 mM phenylmethylsulfonyl fluoride (PMSF)) and placed on ice for 1 h with occasional vortex. Centrifugation followed at 10,000 rpm for 10 min and each cell lysate (50 μg) was subjected to sodium dodecyl sulfate (SDS)-polyacrylamide gel electrophoresis (PAGE) and transferred to nitrocellulose membranes (Pall, New York, NY, USA). Blots were blocked with 5% skim milk in TBST for 1 h at room temperature, then incubated with indicated primary antibodies (Cleaved Caspase-3, Cleaved Caspase-9, Cleaved PARP, Caspase-8, bax, Bcl-2, Mcl-1, survivin, γH2AX, stat3, p-stat3, erk1/2, p-erk1/2, AKT, and p-AKT, 1:2000, Cell Signaling Technology, Danvers, MA, USA) overnight at 4 °C, followed by incubation with anti-rabbit or anti-mouse horseradish peroxidase-conjugated immunoglobulin G (IgG) and visualized with enhanced chemiluminescence.

#### 4.2.8. RT-PCR

Total RNA was extracted from cells treated with indicated concentrations of 12e for 24 h using RNAiso Plus (TaKaRa, Tokyo, Japan). Total RNA (2 µg per sample) was converted to cDNA using 5× all in one RT mastermix (Abmgood, Zhenjiang, China) following the manufacturer’s instructions. Individual reverse transcription reactions of 20 µL were then diluted to 200 µL with ddH_2_O. 20 µL of diluted RT reaction was used for real-time qPCR using the iTaq SYBR Green Supermix with ROX (Bio-Rad, Hercules, CA, USA). The sequences of primers used in real time qPCR reactions were as follows: 5′-TGGCGTAAGATGATGGA-3′ (survivin forward) and 5′-TAGGGACGACGATGAAA-3′ (survivin reverse), 5′-AACACGTACTTGTGCG-3′ (XIAP forward) and 5′-ACTTTGATCTGGCTCA-3′ (XIAP reverse), 5′-AAAGCCTGTCTGCCAAAT-3′ (Mcl-1 forward) and 5′-TATAAACCCACCACTCCC-3′ (Mcl-1 reverse), 5′-GCCTTCTTTGAGTTCG-3′ (Bcl-2 forward) and 5′-CAGCCTCCGTTATCC-3′ (Bcl-2 reverse). β-actin was used as an internal control. Triplicate qPCR reactions were performed for each of the samples. The real-time qPCR condition is 95 °C for 2 min as a pre-denature step, followed by 40 PCR cycles at 95 °C for 10 s, 60 °C for 45 s and 72 °C for 30 s. The data were analyzed using the Applied Biosystems 7300 Real Time PCR System (Thermo Scientific, Waltham, MA, USA) and normalized to β-actin.

#### 4.2.9. DNA Relaxation Assay

DNA relaxation assay was performed as described [40] with slight modification. In brief, a total of 20 µL reaction containing pBR322 DNA (0.5 µg), Topo I (1 U), 10 × DNA Topo I Buffer (2 μL), 0.1% BSA (2 µL) with or without the compound was incubated at 37 °C for 30 min. The reaction was terminated by adding 2 μL of DNA 10 × loading buffer and subjected to electrophoresis in 1% agarose. Then gels were stained with 3× GelRedTM Nucleic Acid Gel Stain and visualized by FluorChem E (Protein Simple, San Jose, CA, USA).

#### 4.2.10. In Vivo Studies

Male BALB/c-nu mice (4 to 6 weeks of age) were purchased from Hunan SJA Laboratory Animal Co., Ltd. (Changsha, China). All animal experiments were approved by the Institutional Animal Care and Use Committee of Ocean University of China and according to standard institutional guidelines. 2 × 10^6^ NCI-H446 or A549 cells were injected subcutaneously into the right flank of nude mice. The tumor volumes of these mice were checked to assess tumor growth. When the tumors reached about 100 mm^3^, the tumor-bearing mice were randomly allocated to various experimental group (*n* = 5, per group), and treated by intragastric administration with FL118, 12e, or vehicle. The formulation for FL118 and 12e in this study used the formulation recipe which contains FL118 or 12e (1 mg/mL), DMSO (5%), Tween-80 (4%), HS-15 (4%), and saline (87%). The corresponding vehicle solution in the basic formulation recipe contains DMSO (5%), Tween-80 (4%), and HS-15 (4%) in saline.

The mice body weights were weighed three times a week. The lengths (L) and widths (W) of tumors were measured three times a week using electronic calipers and the volumes (V) was calculated as following formula: V = (L × W^2^)/2. When the tumor volume reached about 1000 mm^3^, tumors were excised, weighed, ground, and then disrupted on ice for 30 min in loading buffer, and boiled for 10 min. Protein levels were analyzed by western blotting. The antitumor effect of the compounds was determined by inhibition rate (IR), which calculated according to the following equation: IR (%) = (tumor volume of control − tumor volume of compound)/(tumor volume of control-tumor volume of d_0_) × 100%.

#### 4.2.11. Data Analysis

One-way ANOVA with Tukey’s post hoc test was used for statistical analysis of the data, and values were expressed as mean ± SD. Differences of *p* < 0.05 were considered statistically significant.

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
