# Peer review of "Design, Synthesis, and In Vitro/In Vivo Anti-Cancer Activities of Novel (20S)-10,11-Methylenedioxy-Camptothecin Heterocyclic Derivatives"

_ijms, 2020, doi:10.3390/ijms21228495_

Round 1

Reviewer 1 Report

Dear Author,

It is my great pleasure to review your work about novel (20S)-10,11-methylenedioxy-camptothecin heterocyclic derivatives. The study is very thorough, and the manuscript is well drafted. Please find my comments below. Thanks!

  1. Key words: consider a better way to put this key molecule in the key words.
  2. The second molecule: what is purpose of it listed there? Is it a typo of FL118 (isomer mixture)?
  3. Line 104, would not it be better to include FL118 as a positive control in plate clone formation assay?
  4. It was commented in line 276-7 that 12e showed higher proliferation inhibition activity on A549 than on NCI-H446 in vivo. Why? according to line 223-24 and figure 6 Aand B, in vivo inhibition of 12e for NCI-H446 is better than A549.
  5. Line 292-3 mentioned the structure-function-relationship, which not really articulated in the manuscript.
  6. Descriptions about the length of the experiment in line 123 and line 469 are not agree to each other.
  7. Change the font of the Year in the references to BOLD
  8. Check the pdf attachment for other corrections.

Author Response

Dear Editor and Reviewers,

We have taken all comments into careful consideration and reply point by point. Please see the attachment.

Reviewer 2 Report

The manuscript IJMS-960198 describes the synthesis of a series of camptothecin derivatives decorated with ester functions carrying uracil or nitrogen-containing aliphatic cycles, endowed with antiproliferative activity on cancer cell lines.

The authors carried a great effort, particularly in the biological section, but the manuscript needs a careful revision by a native-speaking proof-reader. There are many mistakes, and some statements are incomprehensible.

The paper should be revised also considering the following remarks:

- In the Abstract, compound FL118 was mentioned. For people not familiar with the topic, a definition as:  ..a camptothecin analogue…should be added.

- In the Introduction, I do not understand what primary structure (line 56) of FL118 means. More important, already in this section the authors should describe the rational for their design strategy: the insertion of an ester function and of uracil and heterocyclic moieties on it.

- In the Chemistry, in authors’ opinion, the chemical reactivity of the OH group of FL118 is lower (line 689) with respect to which compound/compounds? Or it is a grammatical mistake?

- In section 2.2, the higher activity and cytotoxicity of new compound 12e with respect to FL118 could be related to its higher solubility?

- In the Discussion, the authors state that it was once believed that intact alpha-hydroxy lactone ring was indispensable for the activity. If I understood well all the sentence, the presence of the hydroxy group on the ring was considered important, and not the dimension of the ring. I do not understand the definition of side effect in this sentence (line 243).

Author Response

(The authors gave the same response as above.)

Reviewer 3 Report

The design of the molecules is rather vague. It just looks like the synthesis of some more derivatives. Could you provide a more elaborated rational for the design? The characterization is well done and delivers similar results than FL118, except for topo I. Why is 12e not active on topo I and how is this linked to your design approach? Do you have any structural explanation for this? Or is this just coincidence? The manuscript would benefit from such a reflection.

Author Response

Dear Reviewer,

Thank you for your  comments concerning our manuscript entitled “Design, synthesis and in vitro/in vivo anti-cancer activities of novel (20S)-10,11-methylenedioxy-camptothecin heterocyclic derivatives”. Those comments are all valuable and very helpful for revising and improving our paper. We have taken these comments into careful account and revised the manuscript and highlighted them accordingly. The responses to the reviewers’ comments are included below.

Point 1: The design of the molecules is rather vague. It just looks like the synthesis of some more derivatives. Could you provide a more elaborated rational for the design? The characterization is well done and delivers similar results than FL118, except for topo I. Why is 12e not active on topo I and how is this linked to your design approach? Do you have any structural explanation for this? Or is this just coincidence? The manuscript would benefit from such a reflection.

Response 1: Thanks for this suggestion. We have designed and synthesized these novel 20-substitued FL118 derivatives based on the Structure-activity relationship of camptothecins, since FL118 and camptothecin have the same core structure. And we have reorganized the design parts in our manuscript. Studies have been demonstrated that the presence of targeted Topo I is not critical for antitumor activity of FL118, although it has the same core structure with camptothecin. In this manuscript, we also got the same result for 12e. We supposed the 10,11-methylenedioxy group of FL118 was a key factor for the different biological mechanisms.